# A Blueprint for Cancer-Related Inflammation and Host Innate Immunity

**DOI:** 10.3390/cells10113211

**Published:** 2021-11-17

**Authors:** Lucia García-López, Isabel Adrados, Dolors Ferres-Marco, Maria Dominguez

**Affiliations:** Mechanisms of Growth Control and Cancer Lab, Instituto de Neurociencias, CSIC-UMH, Avda. Ramón y Cajal s/n, 03550 Sant Joan d’Alacant, Alicante, Spain; dferres@umh.es (D.F.-M.); m.dominguez@umh.es (M.D.)

**Keywords:** tumor–host interactions, inflammation, innate antitumor immunity, diet, autophagy, cachexia, cancer initiation, juvenile, adult, *Drosophila*

## Abstract

Both in situ and allograft models of cancer in juvenile and adult *Drosophila melanogaster* fruit flies offer a powerful means for unravelling cancer gene networks and cancer–host interactions. They can also be used as tools for cost-effective drug discovery and repurposing. Moreover, in situ modeling of emerging tumors makes it possible to address cancer initiating events—a black box in cancer research, tackle the innate antitumor immune responses to incipient preneoplastic cells and recurrent growing tumors, and decipher the initiation and evolution of inflammation. These studies in *Drosophila melanogaster* can serve as a blueprint for studies in more complex organisms and help in the design of mechanism-based therapies for the individualized treatment of cancer diseases in humans. This review focuses on new discoveries in *Drosophila* related to the diverse innate immune responses to cancer-related inflammation and the systemic effects that are so detrimental to the host.

## 1. Introduction

Cancer is a major public health issue that causes close to 10 million deaths every year [1]. The growth of a malignant tumor in a specific organ or tissue often induces its dysfunction, but it is mostly the systemic physiopathological alterations that occur in distal organs that result in devastating outcomes and contribute to the death of the host.

These general effects, which were once thought to be mere metastases, are now known to also be a consequence of secreted proteins and hormones, exosomes, and/or metabolites that enter the bloodstream, affecting different organs and provoking physiological alterations that can result in weight loss, anorexia, fatigue, chronic pain, and other debilitating conditions.

Cell culture and vertebrate models have served us well in understanding the biological mechanisms of cancer development and progression [2]. In addition, xenograft animals have provided important tools for testing and validating targets and therapeutic strategies. Yet cancer initiation, which can only be studied in situ in tumors, the systemic effect of cancer, and systemic anticancer immunity, particularly innate immunity, still remain a black box.

Invertebrate cancer models, specifically of the fruit fly, *Drosophila melanogaster*, have long served as a useful and powerful model system to study how tumors impact systemic host physiology and the diverse strategies that different tumors use to cope and survive under challenging conditions. Here, we review the research that has been conducted in several *Drosophila* model systems on tumor–host interactions, focusing on inflammation and host metabolism in cancer progression and interorgan communication [3].

## 2. Immune Responses: Local and Systemic Inflammation

There is a large amount of evidence revealing the dual nature of the immune system, when defining cancer outcome, in either preventing or promoting tumor growth and metastasis [4,5,6]. However, the study of this duality is enormously challenging due to the complexity of the mammalian immune system [7]. Insects rely on an evolutionarily conserved innate immune system, of which studies have served as a blueprint to identify key aspects of mammalian immunity [8]. These discoveries on immunity and the insight provided into disease mechanisms, as well as the new avenues opened for the development of prevention and therapies against diseases, were awarded the Nobel Prize in 2011.

Much of the work on anticancer immunity has focused on adaptive immune responses, such as T-cell activation. However, inflammation and innate antitumor immunity are also important in the emergence of neoplastic cells and in tumor recurrence, particularly in the step from minimal residual disease to active growing recurrence [9]. Inflammation is an innate defensive reaction in response to harmful stimuli (pathogens or injured tissue) and neoplastic cells. Its main function is to eliminate the initial cause of cell injury, clear out necrotic cells and damaged tissues, and initiate tissue repair [10]. While acute inflammation can be beneficial to protect the host from infections or injuries and preneoplastic cells, this immune response must cease when no longer needed such that chronic inflammation is prevented, as this may foster tumor initiation and metastasis in mammals and in *Drosophila* [11,12].

While the relationship between inflammation and cancer was first exposed in 1863 by Rudolf Virchow [13], the molecular mechanisms by which inflammatory signals help cancer cells to thrive continue to remain a mystery. Virchow hypothesized that cancer originates at sites of chronic inflammation and, nowadays, it is widely assumed that a proinflammatory environment constitutes a risk factor for neoplastic growth in cells that acquire overproliferation capacity [14].

Interestingly, as most *Drosophila* models are based on the development of tumors in situ, this cancer model can be of great value in deciphering the initiating steps that lead from a few neoplastic cells developing into full-blown tumors, for example, how incipient neoplastic cells resist and/or escape attacks of the innate immune cells, and how innate immune cells sense and recognize tumor cells from normal cells. The innate immunity of insects relies on humoral and cellular immune responses to fight microorganisms (bacteria, viruses, and parasites) or injuries (Figure 1). This innate immunity is composed of different organs and tissues, including the fat body, gut, and blood cells [15].

The humoral secretion of antimicrobial peptides (AMPs) from the fat body into the hemolymph has the function of lysing microbes as soon as they reach the epithelial barrier [16]. Some AMPs, specifically drosomycin, have been found to be induced by tumor cells and impact tumor growth [17,18,19].

While mammals have numerous types of immune cells, which participate in the innate and/or adaptive immunity, the cellular immune response of *Drosophila* involves only three types of circulating blood cells, generally referred to as hemocytes, which are myeloid-like immune cells, including plasmatocytes, crystal cells, and lamellocytes [20,21].

Crystal cells and lamellocytes, which constitute approximately five percent of total immune cells, are required for wound healing and melanization, which is the equivalent to the complement system in mammals [22,23]. The rate-limiting enzymes that mediate the melanization process are the *Drosophila* prophenoloxidases (PPOs). Crystal cells, in particular, produce PPO1 and PPO2, which contribute to the bulk of melanization in the hemolymph upon injury. Lamellocytes express PPO3, which is believed to contribute to the encapsulation process against parasitoid wasps [24].

Thus far, only plasmatocytes and crystal cells have been implicated in tumor formation and/or tumor-associated inflammation [4]. Plasmatocytes are macrophage-like cells that mediate the phagocytosis of microorganisms and apoptotic cells upon infection or injury. These cells represent ~90–95% of total hemocytes and have the ability to migrate to the tumor site and infiltrate the tumor while producing a cocktail of proinflammatory cytokines [4,25] and chemokines [4,26] (Figure 1). They can be associated with healthy imaginal discs, referred to as tissue-resident macrophages.

Crystal cells play a key role in innate antitumor immunity via the melanization of abnormal and neoplastic cells, and recent studies have uncovered that active crystal cells overexpress the gene *CG10602* [26], which encodes a leukotriene A4 hydrolase that catalyzes the synthesis of the eicosanoid lipid leukotriene B4 (LTB4) from arachidonic acid. LTB4 is a proinflammatory mediator that is produced by myeloid cells in mammals [27] in response to an inflammatory insult, and we recently demonstrated its role in tumor-related inflammation in *Drosophila Notch*-*Pten* tumors [4], where active lipid proinflammatory mediators are also produced by the tumor cells in *Pten*-deficient cells.

LTB4 is a potent chemokine and attractor of immune cells [27,28] and pharmacological inhibition or genetic inactivation of 5- and 15-lipoxygenases and leukotriene A4 hydrolase, enzymes that catalyze the synthesis of LTB4, suppress immune cell migration in insects [28,29,30] and tumorigenesis and inflammation in insect larvae cancer models [4,31]. LTB4 also stimulates the production of proinflammatory cytokines and mediators such as nitric oxide (NO) [4,32,33]. This can enhance and prolong inflammation, facilitating and promoting the appearance of tumor cells at sites of inflammation, leading to a negative prognosis in cancer. Arachidonic and linoleic acid produce a diversity of eicosanoid and eicosanoid-like lipids that can also have anti-inflammatory roles and, therefore, pharmacological inhibition or stimulation of production of these pro- and anti-inflammatory lipids has great potential as an additional strategy for the treatment of cancer-related inflammation [14,34,35], as seen in studies in *Drosophila* [4,36].

Tumor cells communicate locally, promoting an innate immune response in the area widely known as the tumor microenvironment (TME). This local communication can lead to local inflammation.

### 2.1. Local Proinflammatory Cytokines

One of the most studied cytokines produced by tumors is tumor necrosis factor alpha (TNF-*α*), whose single homologue is *Eiger* (*Egr*) in *Drosophila* [37,38]. TNF-α/Egr is a major proinflammatory cytokine produced within the TME with a dual role as an anti- and protumoral factor [38], although not all tumors are sensitive to Egr [39]. The proinflammatory role is well-conserved in *Drosophila* [40], which has permitted us to better understand this duality. This cytokine has both tumor-intrinsic [41,42] and -extrinsic roles, which we discuss below. Thus, the activation of Egr in nontumor cells exerts an antitumoral effect in some *Drosophila* cancer models, where mutant cells are generated in a wild-type background in imaginal discs. In these contexts, Egr-dependent activation of JNK induces tumor cell death [41,42,43,44]. By contrast, hemocyte-derived Egr has also been shown to promote JNK activation in *Drosophila scrib*/*Ras^V12^* tumors, but in this context, the function of JNK is shifted towards tumor cell proliferation and invasion due to oncogenic cooperation with *Ras^v12^* [45,46]. Interestingly, recent advances in mammals highlight the importance of systemic immunity in response to tumors, although the precise mechanisms remain understudied [47]. The fat body of *Drosophila*, which functions as a liver and adipose tissue, is the main organ involved in humoral immunity. In 2014, the group of Dr. Marcos Vidal reported that epithelial tumors remotely activate the Toll immune signaling pathway in the fat body to trigger a systemic immune response. They suggest that *scrib*/*Ras^V12^* tumor-derived Egr promotes the production of the Toll ligand *Spaetzle* (*spz*) by hemocytes, activating the Toll pathway in the fat body. Toll reciprocally restrains tumor growth by inducing tumor cell death in a non-tissue-autonomous manner [48], but the underlying mechanism by which cell autonomous Toll induces tumor cell death in distant organs remains unknown.

Upds are the *Drosophila* homologs of mammalian interleukins [49]. Recent work highlights the implication of unpaired 2 and 3 (Upd2 and Upd3) cytokines in local responses to tumors. Upd and Upd2 also act as insect leptin-like cytokines [50], which suggests a pleiotropic effect of these cytokines in local and systemic inflammation. Upd3 is produced by different types of tumors [4,51]. In *scrib* mutant cells, secreted Upd3 induces JAK/STAT activation in the fat body and hemocytes, which is required for hemocyte proliferation and subsequent tumor suppression [25]. JAK/STAT signaling can exert protumoral effects in different oncogene contexts [25,49,51]. However, in *scrib*/*Ras^V12^* tumors, Upd3 can activate JAK/STAT signal that cooperate with JNK to promote growth and metastasis, whereas in Notch-dependent tumors, JNK is antitumoral [52]. A similar duality exists in mammalian and human cancers [53,54].

In addition to the production of local proinflammatory cytokines by tumors and the TME, immune cells such as crystal cells also produce chemokines to attract other immune cells to the tumor microenvironment to further elicit immune responses both locally and systemically (Figure 1). As inflammation can exhibit pro- and antitumoral roles, there is an urgent need to understand two intriguing questions: which and how tumor-derived cytokines lead to systemic immune responses and how systemic inflammation ultimately leads to multiorgan failure and the death of the host.

Knowledge and understanding of which specific pathways drive the steps from local to systemic inflammation can help in designing strategies to prevent or ameliorate the systemic effect of cancer that weakens patients.

### 2.2. Systemic Inflammation and Metabolism in Cancer

Inflammation and immune responses are often associated with shifts in metabolism, including changes in tumor cells, the host, and immune cells, the latter referred to using the term immunometabolism [55]. Many solid tumors present infiltrating immune cells and release inflammatory cytokines into surrounding tissues and into the bloodstream, which results in systemic inflammation [56].

Systemic inflammation and proinflammatory processes are linked to poor prognosis in patients with cancer, and are often associated with cancer-associated cachexia (CAC), a multifactorial and multiorgan syndrome characterized by a progressive wasting of skeletal muscle and adipose tissue and apparently associated with increased systemic inflammation [57,58]. However, not all cancers with local or systemic inflammation exhibit a wasting phenotype [59]. Nevertheless, CAC is the most-studied whole-body metabolic syndrome associated with cancer and, therefore, it will be the focus of this review in the following sections. This syndrome is not unique to cancer, and several chronic diseases, such as heart failure, infection, obstructive pulmonary disease, and HIV, also lead to cachexia [60]. Advanced-stage cancer patients show CAC 50% of the time, which has an effect on treatment success and patient survival [61], but it can occur even before cancer is first diagnosed.

Cachexia can be accompanied by cancer-associated anorexia, which is not reversed by increasing food intake [58], resulting in significant weight loss, reduced quality of life, and a shortened lifespan [62]. In fact, approximately 30–80% of cancer patients exhibit weight loss depending on the tumor type [63], and up to 30% of people with advanced-stage cancer die not because of the tumor itself, but because of CAC [60,64,65]. Even within the same cancer type, the host physiology and intrinsic differences in the tumor phenotype can lead to variations in the extent to which patients suffer cachexia [66]. In addition, the severity of cachexia is correlated with increased toxicity resulting from chemotherapy which, in turn, provokes further weight loss [60]. Considering the relevance and heterogeneity of such a syndrome, much significant research in the past two decades has focused on CAC, opening many different avenues and providing information, although its underlying mechanisms are still not completely understood.

One of the most important features of cachexia is chronic systemic inflammation, which induces progressive weight and muscle loss [67]. In mammals, it has been shown that different proinflammatory cytokines produced by immune cells and tumor cells can induce cachexia. These include TNF-α, initially termed “cachectin” [68,69], and interleukin-6 (IL-6) [70].

TNF-α has a direct catabolic effect on skeletal muscle through inducing muscle protein degradation [71,72]. In addition, IL-6 is associated with cachexia in rodent models [73,74,75] and is found in high levels in cachectic patients [74,76]. IL-6 induces suppression of protein synthesis in muscle cells [73,74,75] and also induces lipolysis [77]. In *Drosophila*, it was recently described that JAK/STAT and TNF-α/Egr signaling are elevated in cachectic muscle and promote tissue wasting in a model of *scrib*/*Ras^V12^* tumor-bearing larvae [78], which recapitulates the “high inflammation” that is a hallmark of human cancer cachexia. This study, although it does not demonstrate that TNF-α/Egr is derived from the tumor tissue, constitutes an interesting proof of principle. Here, below, we review the discoveries made in *Drosophila* that have shed light into the mechanisms of cancer cachexia.

#### 2.2.1. Tumor-Secreted Factors Involved in Cachexia in Juvenile *D. melanogaster*

Animals have evolved mechanisms to sense and withdraw from physiological and environmental perturbations to maintain stability and homeostasis [79]. In the fly larvae, perturbed growth, injured tissue, and tumor cells activate the *Drosophila* relaxin peptide Ilp8 (insulin-like peptide 8) [80]. Relaxin peptides belong to the same superfamily as the insulin and insulin-like growth factor (IGF) peptides, but they act through distinct receptors, namely the guanine nucleotide binding protein (G protein)-coupled receptors. Ilp8 is cell-autonomously activated in many if not all tumor cell types in *Drosophila* that develop from diploid cells. Once produced, Ilp8 is then readily secreted in the hemolymph and activates a developmental checkpoint that delays developmental timing and influences global systemic growth (Figure 1) [80].

Ilp8 binds and activates the relaxin receptor Lgr3 (leucine rich repeat-containing G protein-coupled receptor 3) in the central nervous system (CNS) in a still poorly characterized neural circuit that involves insulin-producing cells, juvenile hormone (JH)-regulating neurons, and the prothoracicotropic hormone (PTTH). Tumor-derived Ilp8 activation of Lgr3 in the brain then inhibits production of Ilp3, JH, and PTTH and, consequently, the production of the maturation hormone ecdysone [80,81,82,83,84].

As such, larvae with tumors induced in the imaginal discs, the brain, or blood cells activate Ilp8 and the developmental time checkpoint, delaying sexual maturation and extending the time imaginal discs spend fostering tumor growth and the cachexia-like fat body waste phenotype (Figure 1).

*Ilp8* was independently discovered by two groups in screens to identify candidate genes that mediate tumor-associated developmental delay using oligonucleotide microarrays [80] in the eyeful cancer paradigm [85] and in an unbiased RNAi-based screen [81]. Garelli et al. (2012) also showed that an *Ilp8* allele, *Ilp8^MI00727^*, serves as a powerful tool in cancer studies. *Ilp8^MI00727^* (an eGFP protein trap line) is silenced in normal growing cells or expressed at low levels but becomes strongly activated in a cell-autonomous manner in response to tumor growth, and this was visualized in vivo by the eGFP protein [80]. As *Ilp8* is activated in tumor cells in a nearly universal manner, and in proportion to tumor burden [80,81,86], regardless of the driving oncogene, it serves as a real-time, accurate, in vivo tumor sensor.

Recently, Yeom and colleagues (2021) reported that tumor-secreted Ilp8 also induces anorexia via the Lgr3 receptor in the brain. In this study, they used a *Drosophila* cancer model driven by yki in the adult eyes. They found that the Ilp8–Lgr3 axis is activated in adult Yki-driven tumors and upregulates anorexigenic nucleobinding 1 (NUCB1) and downregulates orexigenic short neuropeptide F (sNPF) and NPF expression in the brain. They also provided evidence that mammalian tumors secrete the relaxin peptide INSL3, which the authors propose is the mammalian homologue of fly Ilp8. Like in flies, INSL3 transcript levels are increased in mice transplanted with tumors, which is also accompanied by the upregulation of anorexigenic signals. Consistent with this, they also found that food intake was significantly reduced in INSL3-injected mice. In human patients with pancreatic cancer, higher serum INSL3/Ilp8 levels are correlated with increased anorexia [61]. These *Drosophila* studies allowed us to identify unsuspected factors in cancer-related cachexia, demonstrating the usefulness of *Drosophila* studies. Cancer-related anorexia often precedes cachexia, and many cancer patients show a loss of appetite before the symptoms of organ wasting appear [58]. Ilp8 was shown to mediate cancer anorexia phenotype, but not the organ-wasting syndrome. Interestingly, earlier studies had revealed other tumor-derived secreted factors that are required for cachexia and, like Ilp8, ultimately converge on systemic insulin signaling [87].

#### 2.2.2. Tumor-Secreted Factors Involved in Cachexia in Adult *D. melanogaster*

The first observations of a wasting phenotype in flies were made by Elisabeth Gateff and Howard A. Schneiderman in 1974. They noticed that flies transplanted with imaginal discs mutant for the tumor suppressor *lethal (2) giant larvae* (*l(2)gl*) develop what they called “the bloating syndrome”. The abdomen of these tumor-bearing flies became swollen and translucent, and the fat body and ovaries degenerated [88]. Strikingly, although this phenotype is very robust, the molecular basis remains unknown.

Decades later, Figueroa-Clarevega and Bilder (2015) found that transplanted tumor eye imaginal discs (*scrib*/*Ras^V12^*) in adult flies can induce cachexia-like phenotypes [89]. They also identified the tumor-secreted factor imaginal morphogenesis protein-Late 2 (ImpL2), a secreted insulin-signaling antagonist that functions by directly binding to *Drosophila* insulin-like peptide 2 (Ilp2) [90]. ImpL2 is the fly homologue of the human insulin growth factor binding protein (IGF binding protein) and drives wasting by reducing insulin signaling in peripheral tissues of tumor-bearing adults. However, tumor-specific inhibition of ImpL2 only partially ameliorates the wasting phenotype, indicating the need to uncover other aspects of tumor–host interactions by means of investigating tumor-derived metabolites. Consistent with those findings, the group of Nobert Perrimon [87] showed that ImpL2 is a cachexic mediator in a different fly tumor model overexpressing *yorkie* (*yki*) in the adult midgut, which leads to wasting of the ovary, fat body, and muscle associated with systemic insulin resistance [87], a feature also reported in human patients and mouse models of cachexia [91,92]. In addition, these gut tumors also perturb whole-body metabolism by increasing hemolymph trehalose and diminishing glycogen and triglyceride storage. Nevertheless, depletion of ImpL2 in these tumors induces a significant but not total rescue of the organ-wasting phenotypes. These results point out the existence of additional mechanisms contributing to cachexia. Moreover, Perrimon’s lab reported in 2019 that yki-induced gut tumors secrete Pvf1, which triggers host Pvr/MEK signaling and wasting of muscles and the fat body [93], suggesting a role for the MEK/ERK pathway in promoting catabolism in peripheral tissues [94,95].

To summarize, at present only a few tumor-secreted factors (ImpL2, Pvf1, and Ilp8) have been identified as cachectic inducers in adult and juvenile stages of *Drosophila* by using different fly tumor models [62,88,90,94], and it is still unknown if other tumor types are also able to induce wasting.

#### 2.2.3. Tumor-Induced Non-Autonomous Autophagy

One of the main mechanisms of tissue degradation under cachexic conditions is autophagy [96,97,98], a housekeeping catabolic process that cleans out damaged macromolecules and defective organelles to provide energy [99]. Intratumor autophagy can have both tumor-suppressing and -promoting roles depending on the context [100,101]. Here, we review the research conducted on non-autonomous autophagy.

Tumor cells in larval wing discs have the capacity to non-autonomously trigger autophagy in the surrounding wild-type cells and affect the microenvironment [102]. However, the tumor-derived factor(s) that drive non-autonomous autophagy remain undefined. Several studies have linked reactive oxygen species (ROS) signaling with autophagy, making it a perfect candidate for further investigation [103].

In *Drosophila*, autophagy has been investigated in the context of the *scrib*/*Ras^V12^* paradigm. In adult flies, these tumors induce a cachexia-like response throughout the entire fly [90]. In larvae, eye-specific *scrib*/*Ras^V12^* malignant tumors release reactive oxygen species (ROS), leading to local and systemic non-cell-autonomous autophagy in gut, muscle, and adipose tissue, with striking effects on tumor growth. Moreover, they demonstrated that inhibition of autophagy in either the tumor microenvironment or peripheral tissues is sufficient to significantly inhibit tumor growth and invasion [104]. In addition, Manent et al. (2017) provided evidence that ROS derived from tumor cells induces non-autonomous autophagy by activating JNK signaling in neighboring cells [102].

Furthermore, it has been proposed that some inflammatory cytokines released by host tissues, such as the fat body or the tumor itself, might have a role in cachexia [105]. For example, another *Drosophila* tumor-secreted factor, Upd2, does not cause anorexia or cachexic phenotype in flies [88,90]. It remains unclear if circulating cytokines are capable of inducing organ wasting in *Drosophila*. However, it has been recently reported that tumor-derived Upd2 (the fly homologue of interleukin-6 (IL-6)) induces non-cell-autonomous autophagy around tumor tissues in the *Drosophila* cancer model of *scrib*/*Ras^V12^* eye imaginal discs [104]. In addition, it has been reported that the expression of Upd/IL-6 cytokines is elevated in *Drosophila* neoplastic tumors [25,85,106]. Interestingly, human cachectogenic cancers are commonly associated with the ability to induce systemic autophagy through the secretion of proinflammatory cytokine IL-6 [98]; hence, research efforts should focus on the autophagic effect of these molecules in distant tissues. *Drosophila* cancer models may represent a suitable option for exploring the interplay between ROS, Upd/IL-6, and non-autonomous autophagy and understand how metabolic changes in the microenvironment and in distal tissues may affect tumor growth and shape responses in the host.

Another type of tissue degradation is lipolysis. In particular, increased lipolysis is the means by which adipose tissue is primarily degraded during cachexia. Studies in humans suggest that loss of fat mass is an early event in the pathogenesis of this condition in humans. This depletion is not due to loss of fat cells (adipocytes) but is attributed to a decrease in lipids stored in these cells, causing them to be smaller [107]. Loss of fat mass is another key feature of CAC, but the mechanism behind this alteration is unknown [108]. Genetic studies in CAC mouse models show that inhibition of lipolysis ameliorates skeletal muscle atrophy [109], so it could be hypothesized that these two tissue-degrading processes may be linked with each other. Although it has been shown that fat body wasting occurs in certain cancer models [88] in *Drosophila*, no findings have yet been published on whether it is also driven by lipolysis in fruit fly. CAC *Drosophila* models are ideal for studying this process in more detail and digging deeper into its potential link to CAC muscle degradation (see Table 1).

## 3. Diet, Inflammation, and Cancer

Recent epidemiological and animal studies suggest that a healthy diet could prevent 1 in 20 cancers [110], but how dietary components influence cancer is still poorly understood. Several *Drosophila* cancer models, including those of juvenile and adult tumors, have been employed to study the relationship between diet and tumorigenesis, revealing that the influence of diet differs depending on the genotype of the tumor itself.

As mentioned earlier, systemic inflammation is associated with poor prognosis in cancer patients and often correlated with cachexia and weight loss. However, obesity increases the risk of developing cancer [111] through mechanisms that include inflammation [112]. Obesity as well as type 2 diabetes are characterized by elevated circulating glucose levels (hyperglycemia) and systemic insulin resistance. *Drosophila* studies use high-sugar diets to mimic the insulin resistance phenotype observed in human conditions [113]. Dr. Ross Cagan’s group showed that in a fly model in which larval imaginal disc cells express activated Ras and Src oncogenes, there is a shift from benign and localized growth to aggressive tumors when subject to a high-sugar diet. Whereas most host tissues fed a high-sugar diet display insulin resistance, these Ras/Src tumors are insulin-sensitive and take up more glucose. This is the consequence of increased expression of insulin receptor (InR) which, in turn, is activated through Wingless (Wg/dWnt) signaling [114]. They next identified Yorkie (*Yki*), an effector of the Hippo pathway, as the primary source of increased Wg expression in diet-enhanced Ras/Src tumors, and perturbations upstream of the Hippo pathway are sufficient to promote Ras/Src tumor growth. They also showed that increased insulin signaling turns on salt-inducible kinases (SIKs), highlighting the relevance of the SIK–Yki–Wg axis in the paradigm of high-sugar diet *src*/*Ras^V12^* tumorigenesis [115].

Interestingly, under a high-sugar diet, src/*Ras^V12^* tumors promote muscle wasting through tumor-derived *branchless* (*bnl*), a *Drosophila* fibroblast growth factor. In this context, muscle breakdown is correlated with the high levels of free circulating amino acids. Intriguingly, in this high-sugar diet context, tumor cells upregulate the levels of a proline transporter. Blocking this transporter reduced tumor growth and, conversely, feeding the larva extra dietary proline was sufficient to trigger malignancy even in the absence of a high-sugar diet [116]. Additionally, it has been reported that wild-type flies fed a high-fat diet upregulate Upd/IL-6, a central inflammatory mediator implicated in many fly tumor phenotypes. These results connect inflammation, a high-fat diet, and obesity, although the precise mechanisms are understudied [117].

Altogether, these insights from whole-animal *Drosophila* models have identified some key aspects related to the connection between oncogenes and diet. Although it remains to be confirmed whether an excess of dietary sugar interacts with cancer genes in humans in a similar way, this opens the possibility of exploiting these pathways in treating obese and cancer patients.

Equally relevant is the study of dietary conditions of nutrient deprivation or restriction. Because growth is an energy-consuming process, energy restriction lowers the growth of many cancers. In *Drosophila*, restriction of dietary protein content significantly extends the lifespan of wild-type animals [118]. However, Dr. Hugo Stocker’s group discovered that protein restriction was sufficient to enhance the proliferative potential of cells lacking the tumor suppressor *Pten* (commonly mutated in many human cancers), promoting tumorigenesis and leading to the death of the host through an unknown non-autonomous mechanism [119].

The same authors reported that in nutrient-deprived conditions, the mild overgrowth of mitotic *Tsc1* or *Tsc2*-mutant clones in imaginal discs is strongly enhanced due to cell hypertrophy [120]. This highlights that the genetic composition of the tumor itself is a critical element that determines which systemic effects will occur in distant tissues and which dietary intervention might be beneficial. In addition, nutrient restriction leads to increased levels of circulating TNF-α/Egr as a consequence of TOR inhibition in the larval fat body, which can ultimately exert pro- or antitumor effects [49,121,122].

All these findings highlight the drastic effects that diet can have on host metabolism, such as inducing inflammation, which is paradoxically associated with both obesity and organ-wasting processes in *Drosophila* cancer models. Research in flies connecting diet with tumorigenesis is particularly relevant to the human condition, since diets high in both sugar and fat are very common in developed societies. However, although the opposite, dietary restriction, is often associated with reduced risk of developing cancer, it might be detrimental in some oncogenic contexts, pointing to the importance of the tumor genetic composition in predicting the effects of diet on cancer outcome. Future research will be required to determine which specific components of dietary interventions are detrimental or protective to a host with cancer, considering that tumor cells have heterogeneous nutritional requirements.

## 4. Concluding Remarks

The precise mechanisms that mediate the crosstalk between tumors and the host are not completely understood. Tumor initiation, progression, and malignant transformation are influenced not only by genetic and epigenetic initiating events and the microenvironment but also by intrinsic and extrinsic factors, including the past experiences of an individual’s immune system and its metabolism, lifestyle, and diet. Much of the work done in the last decade has focused on cancer cells, the microenvironment, cancer metabolic dysfunctions, local inflammation, and how malignant cells sustain growth and overproliferation [2]. However, recent focus is shifting towards understanding tumor–host interactions and the systemic effects related to cancer.

Tumors are highly demanding of nutrients and energy, thereby influencing nutrient availability in their microenvironment [123,124] and, most importantly, they secrete hormones, growth factors, lipids, peptides, cytokines, ROS, and NO that can affect metabolic pathways in distant tissues, leading to the hypothesis that tumors behave as “metabolic dictators” [125]. These mediators indeed alter host innate immunity by directly influencing metabolic pathways and nutrient-storing organs, such as the fat body [123,124,126,127]. Simultaneously, tumor-derived proinflammatory factors can also induce systemic inflammation, which, in turn, provokes metabolic changes that can be detrimental to distant tissues, such as those observed in cachexia.

Extrinsic factors, such as diet, can also impact host metabolism and tumor growth in both positive and negative ways. Furthermore, obesity and the chronic inflammation associated with diet are risk factors associated with many types of tumors, and these conditions have been mimicked in *Drosophila*.

It is intriguing that both obesity and weight loss in cachexia are associated with chronic inflammation and are processes highly related to cancer, although they are also seen in other diseases. It is important to better understand the interactions in cancer-associated cachexia, a complex syndrome in which both tumor and host tissues can secrete factors that result in the impairment of the whole-body metabolism and organ wasting that operates synergistically to promote tumor growth. However, not all tumors induce a cachexic phenotype, which does not rule out the existence of further undiscovered host metabolic alterations other than organ wasting. Hence, the combination of modern high-throughput techniques such as proteomics and metabolomics in characterizing tumors and peripheral tissues are put forth as the key to identifying novel tumor-derived factors with the potential to induce either tissue wasting or other metabolic alterations not related to cachexia.

Another layer of complexity is the crosstalk of the gut and gut microbiota with tumors. Tumors can influence the gut microbiota which, in turn, can influence host metabolism by regulating the release of gut hormones, directly or indirectly affecting tumor growth. Alterations in the gut microbiota could lead to detrimental effects in the host [128]. As *Drosophila* has a simple microbiome composed of 5–20 microbial species, this model can be easily manipulated in the lab, making it a suitable model to explore the complex tumor–microbiome–host interactions and outcomes [129].

The strength of *Drosophila* as a model lies in its powerful genetic toolkit, together with the ease of developing in situ tumors and tumor allografts, which have permitted independent genetic manipulation of tumors and nontumor tissues to unravel tumor–host interactions. More recently, and of great potential, are the generation of *Drosophila* “avatars” for personalized medicine. Given the evolutionary conservation of many key cancer-causing genes, fly avatars can be used as a platform for drug screening and other applications [130].

Thus, *Drosophila* cancer-related inflammation has emerged as a genetic blueprint for understanding the complexity of tumor initiation in the context of a whole organism and the intricate interactions between incipient cancer cells and innate immune cells and inflammation, processes that have, up to now, remained a black box.

## Figures and Tables

**Figure 1 cells-10-03211-f001:**
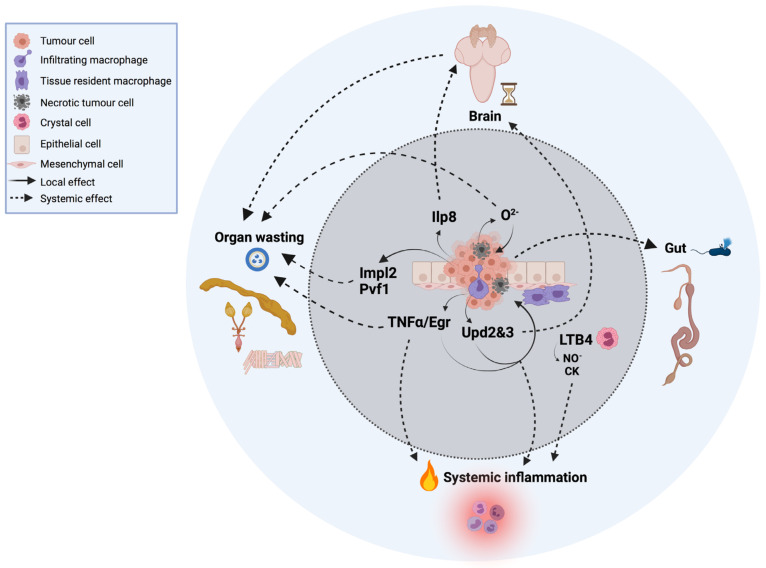
Local and systemic cells and signals in innate immunity and inflammation in cancer in *Drosophila*. For simplicity, the tumor represented in the image is based on a compilation of different oncogenic signatures in which this review focuses on and is explained throughout the text.

**Table 1 cells-10-03211-t001:** Tumor- and tumor microenvironment-secreted factors: local and systemic effects.

*Factor*	*Oncogenic Model*	*Stage*	*Function*	*References*
**TNFα/Egr**	*scrib*/*Ras^V12^* (epithelial tumors)	juvenile	Tissue wasting Systemic inflammation	[79]
**Ilp8**	Universal (discovered in the eyeful tumor metastasis paradigm).	juvenile	Developmental delay Anorexia	[81,86]
**ImpL2**	*scrib*/*Ras^V12^* *yki*	adult	Tissue wasting	[88,90]
**Pvf1**	*Yki* (intestinal tumor)	adult	Muscle and fat body wasting	[94]
**bnl**	*src*/*Ras^V12^* + high-sugar diet	juvenile	Muscle wasting	[109]
**LTB4**	*Delta Pten*-loss (epithelial tumors)	juvenile	Inflammation	[4]
**NO**	*Delta Pten*-loss (epithelial tumors)	juvenile	Inflammation	[4]
**Upd1–Upd3**	Universal	juvenile, adult	Inflammation	[48,55,58,73,77]

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
