# Peer review of "A Blueprint for Cancer-Related Inflammation and Host Innate Immunity"

_cells, 2021, doi:10.3390/cells10113211_

Round 1
Reviewer 1 Report
This manuscript by Garcia et al. reviews recent literature on Drosophila (fruit fly) cancer models. They focus on tumor-host interactions and the immune aspects of this interaction. The authors do a tremendous job of summarizing the latest findings in this hugely active, fast evolving research area. The review is scholarly and well written, with an interesting unifying thesis. I only have minor comments and corrections that I hope the authors find useful.
Line 2: cancer-related
Line 7: Drosophila melanogaster fruit flies
Line 10: black box
Line 36: has long been OR has recently emerged, can’t be both.
Line 46: evolutionarily
Line 58: protect the host from infections
Line 78: AMPs
Line 84: constitute ~5%
Line 97: tissue-resident
Line 105: PTEN deficient cells
Line 119: This sentence feels out of place
Line 123: in… in… in… / Please, rephrase.
Line 129: For discovery of Eiger, Igaki et al., 2002 should be cited in addition to Moreno et al., 2002.
Line 150: distant organs
Line 212: although it does not
Line 216: 3. Cachexia
Line 217: 3.1
Line 238: Ilp8 was discovered independently by two groups IN SCREENS to identify
Line 240: Garelli et al. … / Please, explain better and rephrase.
Line 264: 3.2
Line 296: 3.3
Line 320: However, although if… / Please, rephrase.
Line 323: expression of Upd/IL-6 cytokines is elevated in Drosophila neoplastic tumours / prior work from Tian Xu’s lab could be cited for this.
Line 331: distant tissues
Table 1 is not cited in the text. Is there any reason Upd cytokines are not included in this table? How about MMPs (Uhlirova et al., 2006; Srivastava et al., 2007)?
Line 410: … considering how different dietary components that tumour cells have heterogeneous nutritional requirements. / Please, rephrase.
Line 430: distant tissues
Reviewer 2 Report
This review is very interesting and timely. However, it is very difficult to read and the organizational logic is sometimes quite confusing. Section headings are mis-numbered, and sometime headings for a new section cover a topic that has been discussed in the previous section (e.g., on cachexia). I suggest that the authors revise the ms. with particular attention to logic, flow and grammar and sentence structure, to make it easier for the reader to follow and understand the issues being discussed.
Round 2
Reviewer 2 Report
The authors have made substantial improvements to the manuscript, and it is now very suitable for publication and will be a terrific contribution to the literature.